# Inverse Determination of Acoustic Properties of Acoustic Ventilation Cloth (Woven and Non-Woven) by Particle Swarm Optimization and Estimation of Its Effect on the Frequency Response of the Microspeaker

**Yu-Cheng Liu** [1] , **Suryappa Jayappa Pawar** [2] **and Jin-Huang Huang** [3,*]

[1] Master's Program of Electro-Acoustics, Feng Chia University, No. 100, Wenhwa Rd., Taichung 40724, Taiwan
[2] Department of Applied Mechanics, Motilal Nehru National Institute of Technology Allahabad, Prayagraj 211004, Uttar Pradesh, India
[3] Master's Program of Electro-Acoustics and Department of Mechanical and Computer-Aided Engineering, Feng Chia University, No. 100, Wenhwa Rd., Taichung 40724, Taiwan
* Correspondence: jhhuang@fcu.edu.tw; Tel.: +886-424-517-250 (ext. 3526)

**Abstract:** In this work, microspeaker frequency response was determined based on measurement and simulation. The vents after the rear chamber of the microspeaker were covered with ventilation cloths. Two types of ventilation cloths (acoustic cloths) are commonly used in electroacoustic products, non-woven and woven. Non-woven cloths (5 nos.) consist of irregular meshes and woven cloths (5 nos.) consist of regular meshes. The equivalent circuit model of the microspeaker was formulated by considering five types each of non-woven and woven ventilation cloths. The acoustic properties of the ventilation cloths were estimated from the measured frequency response of the microspeaker and by subsequent use of particle swarm optimization algorithm. Estimated value of the acoustic impedances of the ventilation cloths were used in an equivalent circuit model of the microspeaker for the simulated frequency responses and subsequently compared with the anechoic chamber measurements. Based on the results, the equivalent circuit adequately simulated the measured frequency response of the microspeaker and the estimations of the acoustic impedances of the ventilation cloths were in good agreement with the measured frequency responses of the microspeaker.

**Keywords:** anechoic chamber measurement; equivalent circuit method; frequency response; microspeaker; particle swarm optimization; ventilation cloth

## 1. Introduction

A microspeaker is a tiny electroacoustic actuator that converts electrical signals into sound. With the rapid development of 4C (computer, communication, consumer electronics, and car electronics) products, such as smartphones, notebooks, true wireless stereo earphones, computers, and smart speakers, etc., the microspeaker has become indispensable. In 1924, a patent was awarded to Rice and Kellogg for the moving coil direct radiator loudspeaker. Today, this remains the most significant design in the electroacoustic product (loudspeaker, microspeaker, earphone, headphone, etc.) market. Its miniature version essentially involves the same fundamental components. Loudspeakers include horn loudspeaker, moving coil loudspeaker, electrostatic loudspeaker, ribbon loudspeaker, flat panel loudspeaker, bending wave loudspeaker, and balanced armature loudspeakers, in addition to MEMS speaker, etc. In 1957, the loudspeaker principles and design were carefully addressed in detail by McLean [1], with loudspeaker design criteria have also been established by Chernof [2] in 1957. In 2006, an optical measurement of loudspeaker cone vibration was reported by W. Klippel and Schlechter [3], using the "Laser Triangulation" technique, which is a cost-effective alternative to "Doppler Interferometry" and has been proven to be the most versatile tool for loudspeaker research and industry. Loudspeaker modifications have

been suggested for high fidelity sound reproduction [4–6]. In a significant contribution, Small [7–12] made a significant contribution to this field by elaborating on loudspeaker performance in closed and vented boxes. An equivalent circuit method (ECM) for portable multimedia loudspeaker was discussed by Tashiro et al. [13]. Moreover, Hwang et al. published an extensive report on the microspeaker design and development [14–17]. Acoustic analysis and design of a miniature mobile phone loudspeaker (microspeaker) were carried out by Bai et al. [18,19] using an ECM and finite element method (FEM). Electroacoustic simulation and experimentation were conducted on cellular phone microspeaker have been performed by Huang et al. [20]. To improve the overall sound pressure level (SPL) performance over the mid-frequency spectrum for a microspeaker investigated by Shiah et al. [21] analyzed a few key design parameters in 2008.

In 2003, Moholkar and Warmoeskerken [22] attempted to identify the acoustical characteristics (effects of entrapped air pocket, power consumption of ultrasound horn, transmitted acoustic pressure amplitude, structural and hydrodynamic characteristics, and acoustic impedance) of textile materials using precision woven monofilament fabrics as a model textile. In 2013, Nordgren et al. [23] discovered the influence of anisotropy in open-cell porous materials on the vibroacoustic response. Wintzell [24] extensively documented on acoustic textiles for home wall panels and demonstrated that low-density textile materials are disadvantageous for the sound diffusion. In addition, textiles may be either acoustically absorbent or transparent, depending on the textile build up. The textile structure includes small air pockets that could affect the acoustic properties. Additionally, textiles can be non-woven, woven, or knitted and are normally not homogenous. Various methods are used to make acoustic textiles. In 2013, Ballantyne and Heden [25] utilized an acoustic grille for the loudspeaker cabinet. Tsai et al. [26] have tuned the acoustic impedance of porous materials used in an insert earphone for SPL response based on ECM modeling and measurements to validates and understand the effect of acoustic porous material in 2012. In another study, Liu et al. [27] have investigated the acoustic parameters (air permeability, reflection coefficient, and acoustic impedance) of acoustic cloth used in electroacoustic products in 2014. McIntosh [28] conducted measurements and modeling of circumaural headphones for a desktop computer speaker to investigate the correlation between the measurement and modeling. However, it has been reported that most electro-acoustic devices are designed through "experience" and "trial-and-error" due to the non-availability of acoustic component's impedance, precisely a black cloth (very low acoustic flow resistance) and white cloth (much higher flow resistance) that generally covers the speaker flange. In their work, Klasco and Tatarunis [29] measured the impedance of an acoustical speaker's impedance and highlighted that the acoustic resistance of a meshed cloth could prove to be influential in optimizing the transient response of larger speaker drivers into smaller enclosures. Both woven and non-woven materials result in acoustic impedance. Woven materials are repeatable, uniform, and have less than 10% maximum performance deviation, however, they are costly. In contrast, non-woven materials are cheap but show 30% acoustic resistance variation and are affected by humidity. Acoustic meshes can improve impulse response settling time (damping) and be modeled as a "resistor" across an inductor or capacitor or other tuned circuit. Acoustic meshes control venting, air bearing, damping, acoustic transparency and absorption of dirt, dust, and moisture. Acoustic cloths also provide for the chuffing of the vented pole piece resulting from air velocity turbulence and reduces the volume of air in the vicinity of the vented pole piece to reduce the vent noise and cooling requirements. In 2017, Horoshenkov [30] reviewed the acoustical laboratory methods for determination of porous material's morphological characteristics. Various studies have been carried out on the ECM modeling of the microspeaker. For example, Tashiro et al. [13] have used ECM to study the performance of a flat-panel loudspeaker, generally used for portable multimedia. To improve and control the overall sound-pressure performance of cellular phones, many scholars conducted SPL research on microspeakers of the cellular phones for SPL investigations.

The resistance of the ventilation cloths (papers) to the airflow changes the amount of air passing through it; hence, ventilation cloths can adjust the sound quality. The use of ventilation cloths has been widely adopted in the industry for acoustic purposes, but the knowledge required to properly choose ventilation cloths is not widely available in the literature. As aforementioned, the ventilation cloths are selected based on "experience" and "trial-and-error." The lack of research data on and analyses of various ventilation cloths to determine resistance to the sound propagation limits the development of electroacoustic products (microspeakers, earmuff headphones, earphones, etc.). In this study, the influence of different ventilation cloths on the frequency response of the microspeaker was investigated. An inverse calculation method was used to obtain the acoustic resistance of the ventilation cloths to predict the frequency response curve with a greater accuracy. Particle swarm optimization (PSO) was employed to estimate the acoustic resistance of the ventilation cloths using initially measured frequency response as a target function. Furthermore, the acoustic impedance and frequency response of five different types of woven and five different types of non-woven acoustic ventilation cloths were studied.

## 2. Microspeaker

A schematic diagram of the microspeaker used in this study is shown in Figure 1a; the corresponding front side view, rear side view (without ventilation cloth), and rear side view (with ventilation cloth) are shown in Figure 1b–d, respectively. A typical microspeaker mainly consists of six parts: front cover, under yoke (UY), diaphragm, voice coil (VC), polar piece (PP), and magnet. The diaphragm is the crucial part of the loudspeaker for sound production and is generally made of paper, polymer, metal, fabric, or composite, etc. The microspeaker does not have a spider due to the space and size constraints; hence, the suspension is be taken care of by the diaphragm itself, resulting in the crucial role of diaphragm design. The magnetic loop is comprised of a magnet, an upper PP, and a lower PP; its primary function is to generate a uniform and fixed magnetic field to enable the VC to operate. The vibration thus generated by the diaphragm pushes the surrounding air to generate sound waves. Thus, the sound can be transmitted towards the front and rear sides of the diaphragm. There are 11 vents in the microspeaker casing which allow sound from the rear side of the diaphragm to travel to the rear side of the microspeaker (Figure 1c) via the rear chamber.

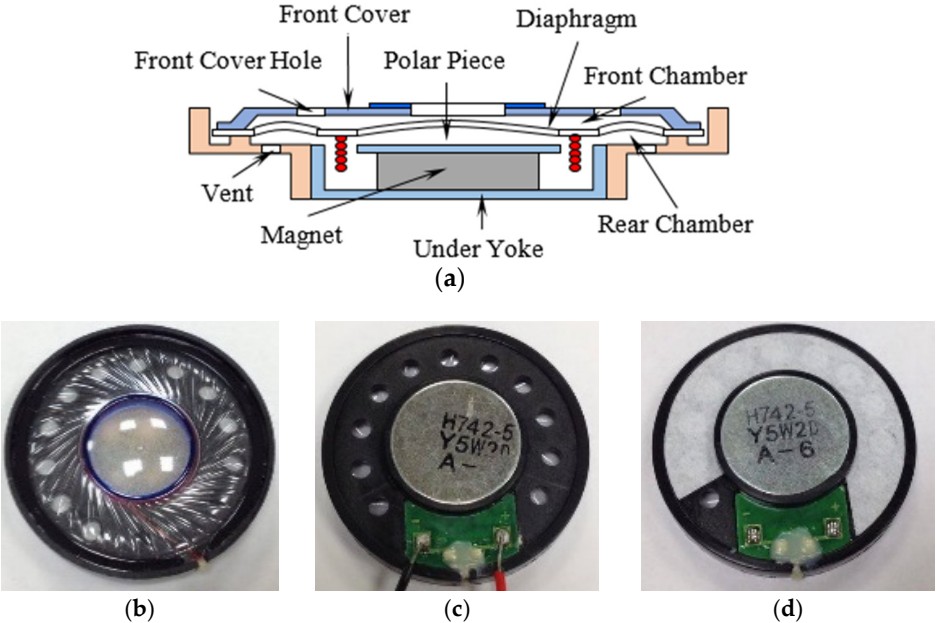

**Figure 1.** (**a**) Microspeaker schematic, (**b**) Front view of microspeaker, (**c**) Rear view (without ventilation cloth) of microspeaker, and (**d**) Rear view (with ventilation cloth) of microspeaker.

The microspeaker investigated in this work was a 40 mm diameter unit (DSH742-005) from Merry Electronic Co., Ltd., Taichung, Taiwan (ROC), with 11 acoustic holes (vents) in the casing towards the rear side of the microspeaker. Table 1 shows the main specifications of this microspeaker. In addition, its fundamental frequency was 80 Hz $\pm$ 20 Hz.

**Table 1.** Specification of microspeaker DSH 742–005.

| Particulars | Details | Remark |
|---|---|---|
| Impedance | 32 ohm $\pm$ 20% | At 1 kHz |
| Sensitivity | 110 $\pm$ 3 dB | At 1 kHz with V (input) 5 mW (400 mV/B&K 4185) |
| Total Harmonic Distortion (THD) | 100 Hz–200 Hz < 25% and 400 Hz–10 kHz < 15% | |
| Fundamental Frequency (fo) | 80 Hz $\pm$ 20 Hz | |
| Rated Input | 5 mW | |
| Maximum Input | 20 mW | |

## 3. Equivalent Circuit of Microspeaker

ECM was used to model the microspeaker when it was placed on an infinite baffle in an anechoic chamber for measurement of frequency response. The detailed ECM of microspeaker is given in our earlier work [31], with the front chamber and holes in the front chamber after the diaphragm and towards the front. In brief, for quick reference, the ECM of the present microspeaker is discussed in brief as follows (Figure 2). The excitation voltage $e_g$ is input to the microspeaker that generates Lorentz force for driving VC. The product $Bl$ is the force factor. The blocked electrical impedance of the VC is represented by $Z_{eb}$ and the mechanical impedance of diaphragm by $Z_m$. The VC drives diaphragm with velocity $v$. The ECM circuit, which is shown in Figure 2, includes two transformers, one towards the front side of the microspeaker's diaphragm and the other towards the rear side of the diaphragm. The area of the diaphragm is given by $A$, which acts as a transduction factor between the mechanical domain of the diaphragm and the associate acoustic domains.

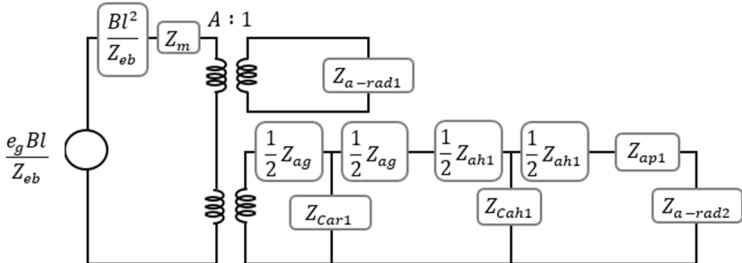

**Figure 2.** Microspeaker equivalent circuit diagram.

As shown in Figure 3, the microspeaker was placed on an infinite barrier; hence, only the sound waves that radiated in the forward direction were received by the microphone, located 10 cm from the microspeaker center. This radiated sound was represented by acoustic resistance ($R_{a\text{-}rad1}$) and acoustic mass ($m_{a\text{-}rad1}$), collectively giving the radiation impedance ($Z_{a\text{-}rad1}$). The sound waves that radiated backward were propagated through the volume of the rear chamber, which was responsible for the cavity compliance impedance $Z_{Car1}$. Additionally, it was estimated that the cavity behind the diaphragm is thin, resulting in a slit-like effect due to acoustic resistance $R_{ag}$ and acoustic mass $m_{ag}$, leading to impedance $Z_{ag}$. Collectively, this cavity was expressed with a T-circuit. Similarly, the sound propagating through the rear openings (vents) of the microspeaker demonstrated acoustic resistance $R_{ah1}$ and acoustic mass $m_{ah1}$, collectively giving acoustic impedance $Z_{ah1}$, which was expressed with a T-circuit. The effect of cavity volume of the rear openings was taken care of by the cavity compliance impedance ($Z_{Cah1}$) in the T-circuit. A ventilation

cloth covered the rear openings, and their combined acoustic impedance was estimated as $Z_{ap1}$. Finally, sound that radiated to the external environment was represented as a parallel combination of acoustic radiation resistance $R_{a\text{-}rad2}$ and acoustic radiation mass $m_{a\text{-}rad2}$ resulting in acoustic radiation impedance $Z_{a\text{-}rad2}$.

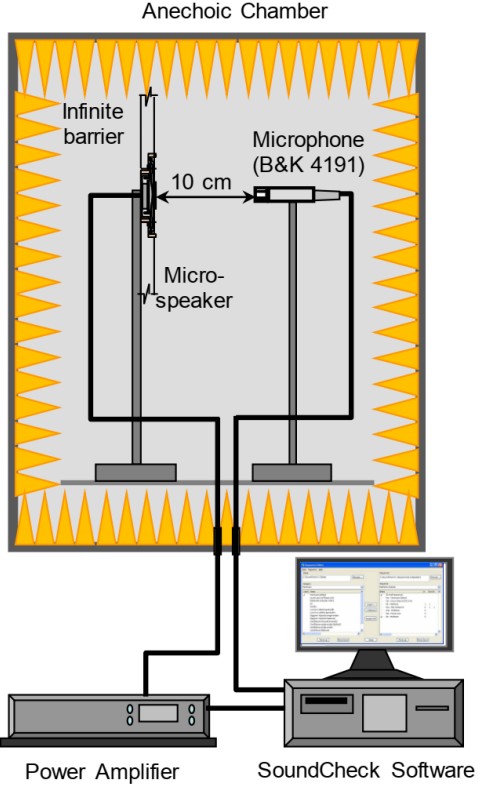

**Figure 3.** Schematic diagram of microspeaker on the barrier in an anechoic chamber.

Our group has reported detailed ECM modeling of loudspeaker (woofer) [32], microspeaker [31], earphones [33,34], and headphone [35]. The mathematical formulation used in the ECM of the current microspeaker is given below in brief:

$$
\begin{aligned}
Z_m &= R_m + j\omega m_m + \frac{1}{j\omega C_m} \\
Z_{eb} &= R_e + j\omega L_e + \frac{R_2 \cdot j\omega L_2}{R_2 + j\omega L_2} \\
Z_{ag} &= R_{ag} + j\omega m_{ag} \\
Z_{Car1} &= \frac{1}{j\omega C_{Car1}}, Z_{Cah1} = \frac{1}{j\omega C_{Cah1}} \\
Z_{ah1} &= \frac{R_{ah1} + j\omega m_{ah1}}{N_{ah1}}, Z_{ap1} = \frac{R_{ap1}}{N_{ap1}} \\
Z_{a-rad1} &= \left(\frac{1}{R_{a-rad1}} + \frac{1}{j\omega m_{a-rad1}}\right)^{-1} \\
Z_{a-rad2} &= \left(\frac{1}{R_{a-rad2}} + \frac{1}{j\omega m_{a-rad2}}\right)^{-1}
\end{aligned}
\tag{1}
$$

All symbols have their usual meaning. $Z_m$ is the mechanical impedance of the diaphragm (assuming it as a simple spring-mass-dashpot system), $R_m$ is the mechanical resistance, $m_m$ is the mechanical mass, and $C_m$ is the mechanical compliance of the diaphragm. The blocked electrical impedance, $Z_{eb}$ is modeled as a series combination of the electrical inductance of VC ($L_e$) and electrical resistance of VC ($R_e$), along with the addition of a parallel combination of resistance due to eddy current ($R_2$) and inductance induced at high frequency ($L_2$). The acoustic impedance of the small duct/port/tube along with the cavity is modeled analogously to the "Helmholtz Resonator" as a series combination of acoustic mass ($m_a$) and acoustic resistance ($R_a$). Thus, $Z_{ag}$ is the acoustic impedance

of the air gap below the diaphragm, and $Z_{ah1}$ is the effective acoustic impedance of the vent, such that $N_{ah1}$ is the number of vents. The acoustic impedance of the cavity ($Z_{Ca}$) is calculated by the compliance of cavity ($C_a$), volume of cavity ($V_c$), density of air ($\rho_0$), and speed of sound in air ($c$). Thus, $Z_{Car1}$ represents the acoustic impedance of cavity below the diaphragm and $Z_{Cah1}$ represents the acoustic impedance cavity due to all vents. Finally, the acoustic cloth covered all vents, so $Z_{ap1}$ represents the acoustic impedance of acoustic cloth, by considering the number of vents ($N_{ah1}$). For more details, readers can refer to our group's earlier works [31–35]. Further, ECM circuit has to be simplified by removing transformers to form a typical three loop circuit to solve it according to Kirchhoff's voltage law. Loops are driven by velocities $v_1$, $v_2$, and $v_3$, respectively, and can be represented by

$$
\begin{bmatrix} \frac{e_g Bl}{Z_{eb}} \\ 0 \\ 0 \end{bmatrix} = \mathbf{Z} \cdot \begin{bmatrix} v_1 \\ v_2 \\ v_3 \end{bmatrix}
\tag{2}
$$

where the impedance matrix $\mathbf{Z}$ is given by

$$
\mathbf{Z} = \begin{bmatrix} \hat{B} & -A^2 Z_{Car1} & 0 \\ -A^2 Z_{Car1} & \hat{C} & -A^2 Z_{Cah1} \\ 0 & -A^2 Z_{Cah1} & \hat{D} \end{bmatrix}
\tag{3}
$$

and

$$
\begin{aligned}
\hat{B} &= \frac{(Bl)^2}{Z_{eb}} + Z_m + A^2 \left( Z_{a-rad1} + \tfrac{1}{2} Z_{ag} + Z_{Car1} \right) \\
\hat{C} &= A^2 \left( Z_{Car1} + \tfrac{1}{2} Z_{ag} + \tfrac{1}{2} Z_{ah1} + Z_{Cah1} \right) \\
\hat{D} &= A^2 \left( Z_{Cah1} + \tfrac{1}{2} Z_{ah1} + Z_{ap1} + Z_{a-rad2} \right)
\end{aligned}
\tag{4}
$$

The diaphragm velocity $v_1$ can be obtained by

$$
v_1 = \begin{bmatrix} 1 & 0 & 0 \end{bmatrix} \cdot \mathbf{Z}^{-1} \cdot \begin{bmatrix} 1 \\ 0 \\ 0 \end{bmatrix} \cdot \left( \frac{e_g Bl}{Z_{eb}} \right)
\tag{5}
$$

Finally, the sound pressure $p$ can be obtained by

$$
p(r) = -j\rho_0 \omega A v_1 \left( \frac{e^{-jkr}}{2\pi r} \right)
\tag{6}
$$

where $r$ (=10 cm, see Figure 3) is the distance of the microphone from the microspeaker. Finally, the sound pressure level of the microspeaker is the pressure level of the sound measured in decibels (dB) and abbreviated dBSPL. On axis SPL can be obtained by

$$
SPL = 20 \log_{10} \left( \frac{p(r)}{p_{ref}} \right)
\tag{7}
$$

where $p_{ref}$ is the reference sound pressure in air, $2 \times 10^{-5}$ N/m$^2$ (0.00002 Pa). The $p_{ref}$ is the lowest sound pressure possible for human hearing.

## 4. Inverse Calculation for Acoustic Impedance of Ventilation Cloth

The acoustic impedance of the ventilation cloth placed over vents is not easy to obtain, and, hence, has mostly been neglected in earlier research. Moreover, it may compromise the accuracy of the ECM circuit. In this study, an estimation of the acoustic impedance of the ventilation cloth was attempted. Due to the porous nature and thickness of the ventilation cloth, it was assumed that the contribution to acoustic impedance ($Z_{ap}$) was due to acoustic resistance ($R_{ap}$) only and the effect of acoustic mass ($m_{ap}$) was negligible.

The PSO algorithm was used to set the SPL measurement curve as the target function, and the acoustic resistance of the ventilation cloth was obtained by inverse calculation. The PSO algorithm was proposed by Kenned and Eberhart [36,37] in 1995. It was originally conceived to imitate the movement of birds or fishes.

The transmission behavior of natural swarms or flocks with a group of particles dropped in the search space has been reported [38]. The PSO optimization revolves around two primary component methodologies. It is understandable due to its ties to artificial life and bird flocking (swarming). It is also related to evolutionary computation and links both genetic algorithms and evolutionary programming. The PSO comprises a straightforward concept that can be executed in a few lines of computer code, requires only primitive mathematical operators, and is computationally inexpensive (in terms of memory requirement and speed). The implementation of PSO relies on two things, nearest-neighbor velocity matching and craziness. Random initialization of the population of birds is performed on a torus pixel grid with a position and X and Y velocities. At each iteration, for each bird, with other agent as its nearest neighbor, determinations are made and X and Y velocities are assigned to the bird in focus. This simple rule creates synchrony of movement. However, the flock quickly settles on a unanimous, unchanging direction. Therefore, a stochastic variable called craziness has been introduced. Each iteration adds some change to randomly chosen X and Y velocities. Such factors introduce enough variation into the system to provide an exciting and "lifelike" appearance to the simulation, while at the same time being wholly artificial. Thus, using the concept of information sharing in a biological society to provides opportunities for the particles in a group to communicate with each other and exchange information in the search for a more effective solution for finding the system optimization parameters. In recent years, PSO has been widely used in acoustics including automatic adjustment of off-the-shelf reverberation effects [39], comparison of optimization methods for compression driver design [40], optimization of micro-perforated sound absorber [41], and optimization of acoustic filters [42].

In this work, the parameter to be solved was treated as a particle swarm, and its initial position was $x$. An objective function $J(x, f)$ was defined to derive the best position $x\_opt$ (best solution) of this particle swarm, as follows:

$$J(x, f) = \sum_{f=20\text{Hz}}^{f=20\text{kHz}} (SPL_m(f) - SPL_s(x, f))^2 \tag{8}$$

The $SPL_m$ is the measured sound pressure level, and $SPL_s$ is the simulated sound pressure level (from the ECM model) of the microspeaker. When the objective function $J(x, f)$ reached the minimum value during the optimization, the simulated SPL approaches the measured SPL. This indicates that the unknown parameter $x$ in the objective function has converged to the optimal solution $x\_opt$. The particle positions and velocities have to be updated continuously during particle movement as follows:

$$x_i^{k+1} = x_i^k + v_i^{k+1} \tag{9}$$

$$v_i^{k+1} = \omega v_i^k + c_1 r_1^k \left( p_i^{k,best} - x_i^k \right) + c_2 r_2^k \left( g_i^{k,best} - x_i^k \right) \tag{10}$$

where $x_i^{k+1}$ is the position of the $i$th particle at the $(k + 1)$th iteration (new position), $x_i^k$ is the position at the $k$th iteration (original position), $v_i^{k+1}$ is a new velocity, $v_i^k$ is the original velocity, $\omega$ is the inertia weight, $c_1$ is the empirical weight of the particle, $c_2$ is the empirical weight of the group, and $r_1^k$ and $r_2^k$ are the random variables between 0 and 1. The random variables change randomly with each iteration, representing the random directions of particles and groups, respectively. Moreover, $p_i^{k,best}$ is the best solution of $i$th particle at

the $k$th iteration, and $g_i^{k,best}$ is the $k$th best solution of the group. The best solution can be judged by the following formula

$$
\begin{pmatrix}
g_i^{k,best} = p_i^{k,best} & J\left(p_i^{k,best}, f\right) < J\left(g_i^{k,best}, f\right) \\
p_i^{k,best} = x_i^k & J\left(x_i^k, f\right) \quad < J\left(p_i^{k,best}, f\right)
\end{pmatrix}
\tag{11}
$$

The $R_{ap}$ of 10 commonly used ventilation cloths were estimated their $R_{ap}$ in this work. These ventilation cloths were sub grouped into two categories viz. (1) Non-woven (NW-1 to NW-5, varying in thicknesses) and (b) Meshed (M-250 to M-450, varying in number of meshes). The details of all 10 ventilation cloths are provided in Tables 2 and 3 (non-woven and meshed ventilation cloths, respectively).

**Table 2.** Detailed specifications of non-woven ventilation cloths of irregular meshes.

| S. N. | Designation | Thickness (m) |
|---|---|---|
| 1 | NW-1 | $5 \times 10^{-5}$ |
| 2 | NW-2 | $7 \times 10^{-5}$ |
| 3 | NW-3 | $9 \times 10^{-5}$ |
| 4 | NW-4 | $13 \times 10^{-5}$ |
| 5 | NW-5 | $16 \times 10^{-5}$ |

**Table 3.** Detailed specifications of meshed ventilation cloths with thickness of $8 \times 10^{-5}$ m.

| S. N. | Designation | Number of Meshes (N) |
|---|---|---|
| 1 | M-250 | 250 |
| 2 | M-300 | 300 |
| 3 | M-350 | 350 |
| 4 | M-400 | 400 |
| 5 | M-450 | 450 |

As aforementioned, the acoustic resistances of all ventilation cloths were estimated by inverse calculation. The ventilation cloths were glued over the vents to the rear side of the microspeaker (Figure 1d), and the frequency response curves were obtained by anechoic chamber measurements. The measured curves were set as a target function in the PSO implementation to estimate the acoustic resistance. The built-in function "particleswarm" in MATLAB software [43] was used to execute the PSO algorithm. The syntax of the MATLAB "particleswarm" in-built function was "$x$ = particleswarm(fun,nvars,lb,ub,options)" to attempts to find a vector "$x$" that achieves a local minimum of "fun" for the given number of design variables "nvars". The "lb" and "ub" are the set of lower (0.01) and upper bounds (20) on the design variables ($x$) to find a solution in the range (lb $\leq x \leq$ ub). "Options" include the default optimization parameters. The PSO consists of seven steps. Step 1 confirms the known parameters (T/S parameters) and target curve ($SPL_m$). Step 2 establishes an ECM model and defines a "particle swarm" (parameter) to be optimized in the model. Step 3 uses a random number function to initialize the particle's position and velocity (say $k$ = 0, for the first iteration). The particle solution $x_i^{k+1}$ is substituted into the ECM in Step 4 to calculate $SPL_s$. Further, the objective function is used to evaluate $p_i^{k,best}$. Step 5 updates the velocity and position of each particle group by updating the particle's velocity $v_i^{k+1}$ and substituting it to obtain updated position $x_i^{k+1}$. In Step 6, the updated position $x_i^{k+1}$ is used to obtain $g_i^{k,best}$ and $p_i^{k,best}$. Step 7 tests the convergence. If the convergence is derived, then the best particle swarm solution is provided. Otherwise, the process is repeated from Step 4. Thus, the best value of acoustic resistances of all ventilation cloths is determined separately.

In this work, the number of iterations was used as the condition for the termination of the calculations. The upper and lower boundaries of 10 to 50 times, number of particles of 20 to 50, number of iterations of 50 to 200, inertia weight $\omega$ of 0.729, and acceleration constants

of 1.494 were set during PSO execution. All the PSO parameters are were based on [44]. Since the PSO algorithm is a random search algorithm, the results of each operation are different, so for the stability of result, multiple calculations must be made to obtain stable and convergent results. Hence, the coefficient of variation was obtained to measure the degree of dispersion of the probability distribution. The closer the coefficient of variation is to 1, the greater the degree of dispersion and stability of results. Thus, the acoustic resistances of all ventilation cloths were obtained for use in ECM simulation.

## 5. Measurements

The KLIPPEL measurement system was used to measure the T/S parameters of microspeaker DSH742-005 by adopting the laser vibrometry principle and standard procedures. The T/S parameters are a set of electromechanical parameters of the loudspeaker driver that define the actual low-frequency performance of the loudspeaker. Loudspeaker manufacturers provide these parameters in specification sheets to aid designers in selecting an appropriate loudspeaker. With these parameters, the designers can simulate the diaphragm's position, velocity, and acceleration. In addition, the input impedance and the sound output of a system comprising a loudspeaker and an enclosure can also be simulated with these parameters. The effective diaphragm area of microspeaker at different frequencies can also be obtained with the SCN module in the KLIPPEL measurement system (Figure 4). Measurement of the frequency response of the microspeaker was done in an anechoic chamber (free space—600 cm (L) × 600 cm (W) × 340 cm (H), cutoff frequency ≦ 100 Hz), as shown in Figure 3. The microspeaker was fixed on the barrier to avoid interference between sound propagated by the microspeaker from front and rear sides. The B&K measurement microphone (B&K 4191). It is an externally polarized 1/2 inch free-field microphone suitable for measurement at 3.15 Hz–40 kHz. It is used for high-precision sound measurements and must be used with a classical preamplifier. The B&K 4191 is complies with IEC 61672 class 1. Initial measurements were carried out to obtain frequency responses of the microspeaker. Next, measurements were carried out to obtain the frequency responses of the microspeaker with different ventilation cloths, as shown in Figure 1d. These frequency responses were used to estimate the effects of different ventilation cloths on the frequency responses for the inverse calculation of acoustic resistances of different ventilation cloths for use in the ECM simulation.

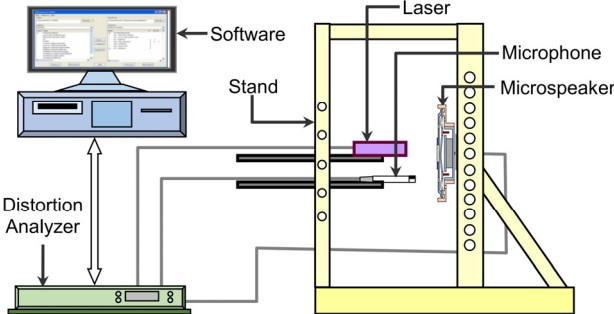

**Figure 4.** Schematic diagram of T/S parameter measurements of microspeaker using KLIPPEL measurement system.

## 6. Results and Discussion

First, the T/S parameters of microspeaker DSH742-005 have been obtained from the KLIPPEL measurement system (Figure 4) using laser vibrometry and are given in Table 4. These parameters define the specified low-frequency performance of a microspeaker. Other important parameters are also shown in Table 4.

**Table 4.** Thiele/Small (T/S) parameters of DSH742-005.

| Symbol | Name | Value |
|:---:|:---:|:---:|
| $R_e$ | Electrical voice coil resistance at DC (Ohm) | 30.81 |
| $R_2$ | Electrical resistance due to eddy current losses (Ohm) | 2.16 |
| $L_e$ | Frequency independent part of voice coil inductance (H) | $1.35 \times 10^{-4}$ |
| $L_2$ | Para-inductance of voice coil (H) | $1.08 \times 10^{-4}$ |
| $M_{ms}$ | Mechanical mass of voice coil and diaphragm without air load (kg) | $1.01 \times 10^{-4}$ |
| $R_{ms}$ | Mechanical resistance of total-driver losses (N·s/m) | $2.2 \times 10^{-2}$ |
| $C_{ms}$ | Mechanical compliance of driver suspension (m/N) | $1.0282 \times 10^{-2}$ |
| $Bl$ | Force factor (Bl) (T·m) | 2.467 |
| $S_d$ | Diaphragm area (m$^2$) | $9.62 \times 10^{-4}$ |

The frequency response curve of the microspeaker without ventilation cloth are shown in Figure 5. The simulated and measured curves are in agreement before the second resonance. The frequency response rises monotonously until the fundamental frequency and becomes flat afterwards until the next resonance frequency. The obvious error between frequency response is at the peak and valley, 4~5 kHz. This difference may arise from the impedance value of the rear openings (vents) of the microspeaker. It could also be due to the mode splitting of the diaphragm at high frequencies. Overall, based on the results, one can conclude that the ECM model of the microspeaker has accurately estimates the preliminary prediction of the characteristics of the microspeaker's frequency response. Thus, the validity of the proposed ECM circuit (Figure 2) has been established, which leads to further was investigated.

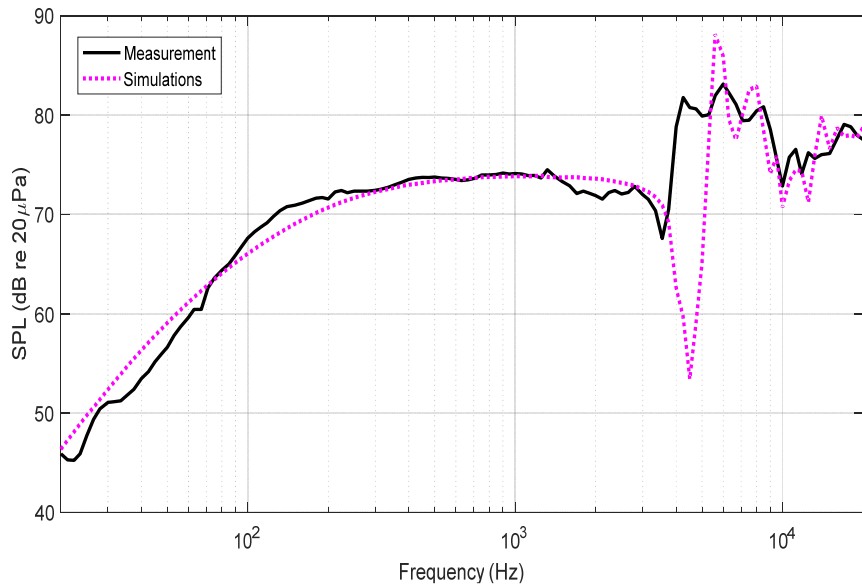

**Figure 5.** Simulated and measured of the frequency response curves of the microspeaker without ventilation cloth.

The initial results of the PSO algorithm implementation in terms of the number of particles and the number of iterations, are shown in Table 5. The coefficients of variation after 30 numbers of particles were all close to 0, indicating that the set of parameters exhibits stable convergence. With selection of the combination of 30 particles and 50 iterations, the result of the upper and lower boundaries are shown in Table 6. Stable and convergent results were obtained in the boundary range of 10 times to 40 times. However the boundary range of 50 times was too wide, resulting in error. Therefore, 10 times boundary range was decided on for the inverse calculation of ventilation cloth parameters. The acoustic resistances of each type of ventilation cloth are shown in Table 7.

**Table 5.** Optimization of acoustic resistance of ventilation cloths, particle number, and coefficient of variation (Remarks: The upper and lower boundaries are in the range of 10 times the original parameter value (10~0.1)).

| Ventilation Cloths | Number of Iterations | | | |
|---|---|---|---|---|
| | **50** | **100** | **150** | **200** |
| **Number of Particles** | Coefficient of Variation (Standard Deviation/Fit Value) | | | |
| 20 | 0.159409265 | 0.159409265 | 0.159409265 | 0.159409265 |
| 30 | $5.05 \times 10^{-9}$ | $4.71 \times 10^{-9}$ | $4.71 \times 10^{-9}$ | $4.71 \times 10^{-9}$ |
| 40 | $5.60 \times 10^{-13}$ | $6.54 \times 10^{-12}$ | $6.57 \times 10^{-11}$ | $6.57 \times 10^{-11}$ |
| 50 | $5.87 \times 10^{-12}$ | $2.63 \times 10^{-13}$ | $3.40 \times 10^{-15}$ | $6.41 \times 10^{-15}$ |

**Table 6.** Optimization of acoustic resistance of ventilation cloths, upper and lower boundary tests (Remarks: 30 particles, 50 iterations).

| Upper and Lower Boundary | 10 Times | 20 Times | 30 Times | 40 Times | 50 Times |
|---|---|---|---|---|---|
| Coefficient of variation | $5.05 \times 10^{-9}$ | $1.87 \times 10^{-11}$ | $1.36 \times 10^{-10}$ | $2.37 \times 10^{-9}$ | 0.285395 |

**Table 7.** Optimized values of acoustic resistance of various ventilation cloths.

| Ventilation Cloths | Initial Value "a" (N·s/m$^5$) | Upper and Lower Boundary |
|---|---|---|
| $R_{ap}$ | $2.0 \times 10^7$ | $(10 \times a)$~$(0.1 \times a)$ |
| **Ventilation Cloths** | **Best Value (N·s/m$^5$)** | **Variation (%)** |
| $NW\text{-}1\_R_{ap}$ | $4.22701 \times 10^6$ | 78.8645 |
| $NW\text{-}2\_R_{ap}$ | $7.17173 \times 10^6$ | 64.1413 |
| $NW\text{-}3\_R_{ap}$ | $1.04271 \times 10^7$ | 47.8641 |
| $NW\text{-}4\_R_{ap}$ | $1.71071 \times 10^7$ | 14.4643 |
| $NW\text{-}5\_R_{ap}$ | $1.93987 \times 10^7$ | 3.0064 |
| $M\text{-}250\_R_{ap}$ | $3.54943 \times 10^6$ | 82.2528 |
| $M\text{-}300\_R_{ap}$ | $4.27662 \times 10^6$ | 78.6169 |
| $M\text{-}350\_R_{ap}$ | $5.94194 \times 10^6$ | 70.2903 |
| $M\text{-}400\_R_{ap}$ | $1.20654 \times 10^7$ | 39.6726 |
| $M\text{-}450\_R_{ap}$ | $5.25710 \times 10^6$ | 73.7095 |

Figures 6 and 7 present the measured and simulated results of the non-woven ventilation cloths, respectively. It can be observed from the measured curves that the influence of the type of ventilation cloth is mainly below 1 kHz. For comparison, the frequency response of microspeaker without ventilation cloth is also given. Below 1 kHz, the highest curve was for microspeaker without ventilation cloth and the remaining curves were progressively lowers as thickness of non-woven cloth increased. Moreover, the remaining curves tended to became relatively straight, facilitating ease of listening. The measurement curves with non-woven ventilation cloth exhibited flat frequency response after 1 kHz until the start of furious resonances. The sharp dip in the response without ventilation cloth was not evident among the remaining responses, ensuring the extension of the audible range to the microspeaker. The simulated curves (Figure 7) also showed similar trends to those of measured curves, thus confirming and validating the ECM model.

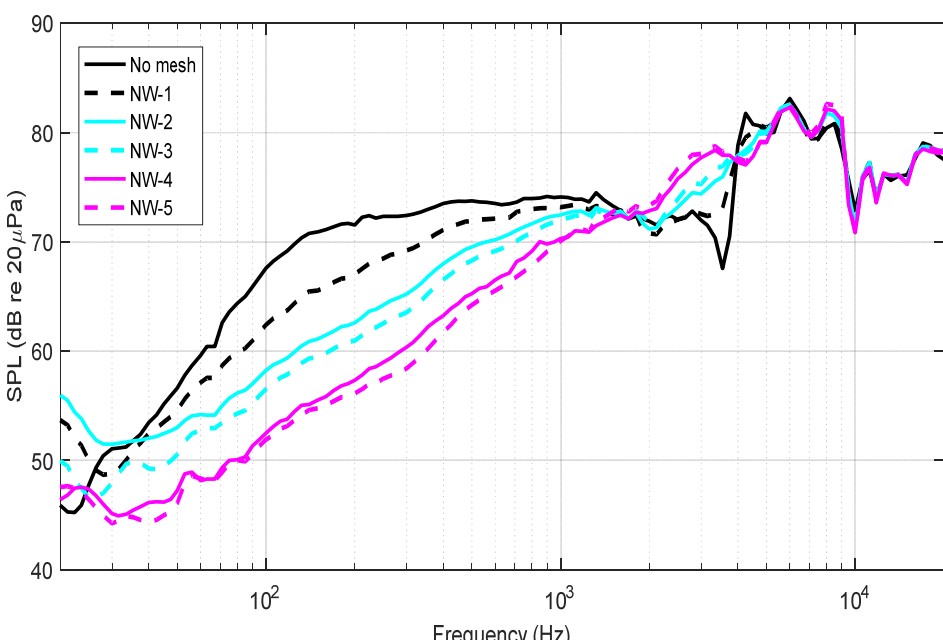

**Figure 6.** Measured frequency responses of microspeaker without and with non-woven ventilation cloths (NW-1 to NW-5).

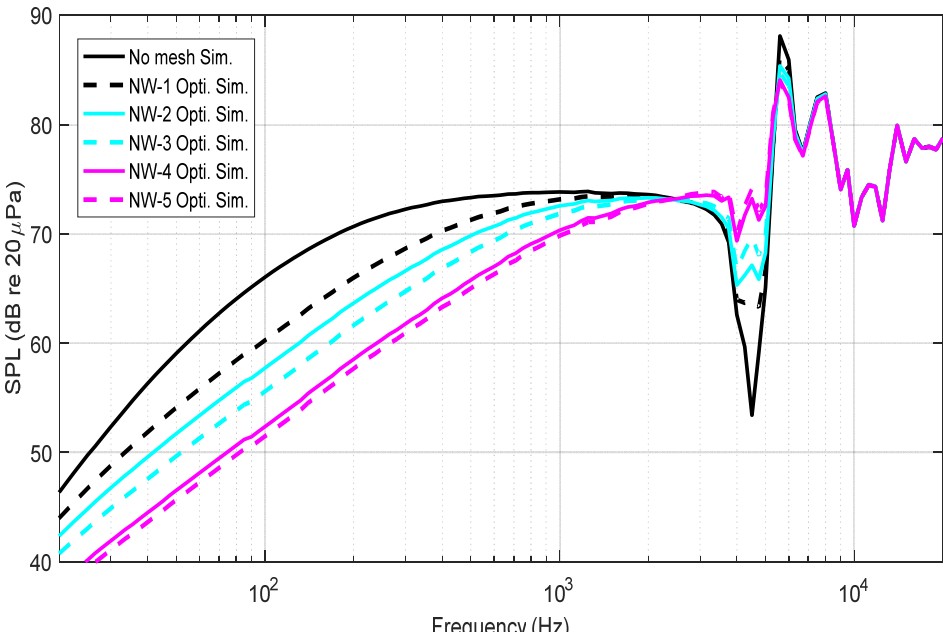

**Figure 7.** Simulated frequency responses of microspeaker without and with non-woven ventilation cloths (NW-1 to NW-5).

Figures 8 and 9 show the measured and simulated results of the meshed ventilation cloths, respectively. The measured responses of microspeaker with and without ventilation cloths show a definite effect. As aforementioned, the measured curves were influenced by the meshed ventilation cloths at mainly below 1 kHz. Moreover, as the number of meshes increased, the curves became lower and straighter below the fundamental resonances. The frequency range after 1 kHz was flat until the start of spurious resonance.

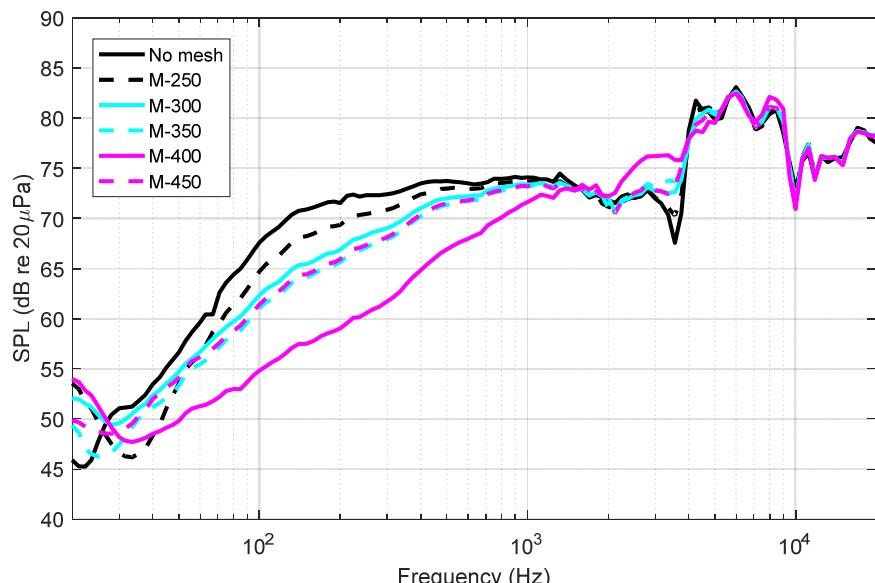

**Figure 8.** Measured frequency responses of microspeaker without and with meshed ventilation cloths (M-250 to M-450).

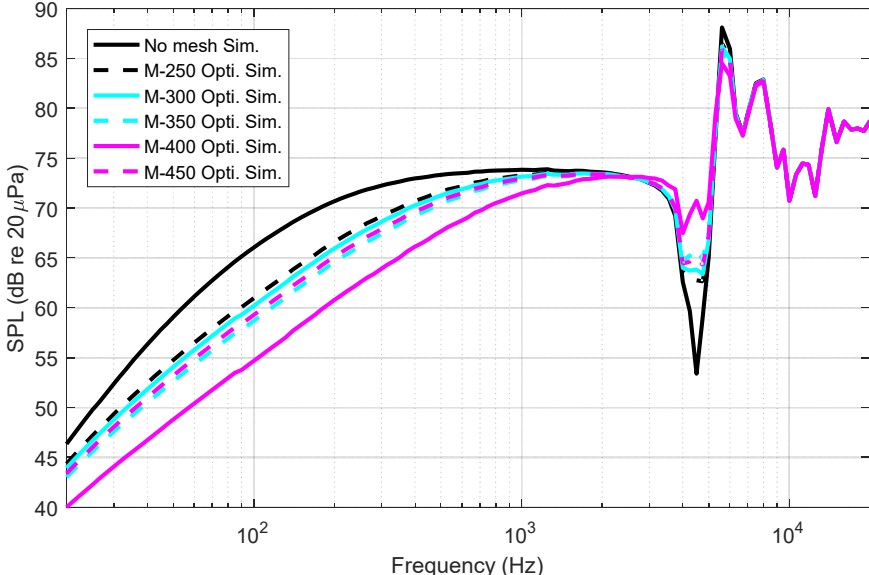

**Figure 9.** Simulated frequency responses of microspeaker without and with meshed ventilation cloths (M-250 to M-450).

Figures 6 and 7 show the measured and simulated frequency responses of microspeaker without and with non-woven ventilation cloths (NW-1 to NW-5). Additionally, Figure 10a,b illustrate the comparison of measured and simulated frequency responses of microspeaker without and with non-woven ventilation cloths (NW-1 to NW-5). The measured and simulated responses show a good match, indicating that the ECM model can accurately predict the frequency response of microspeakers.

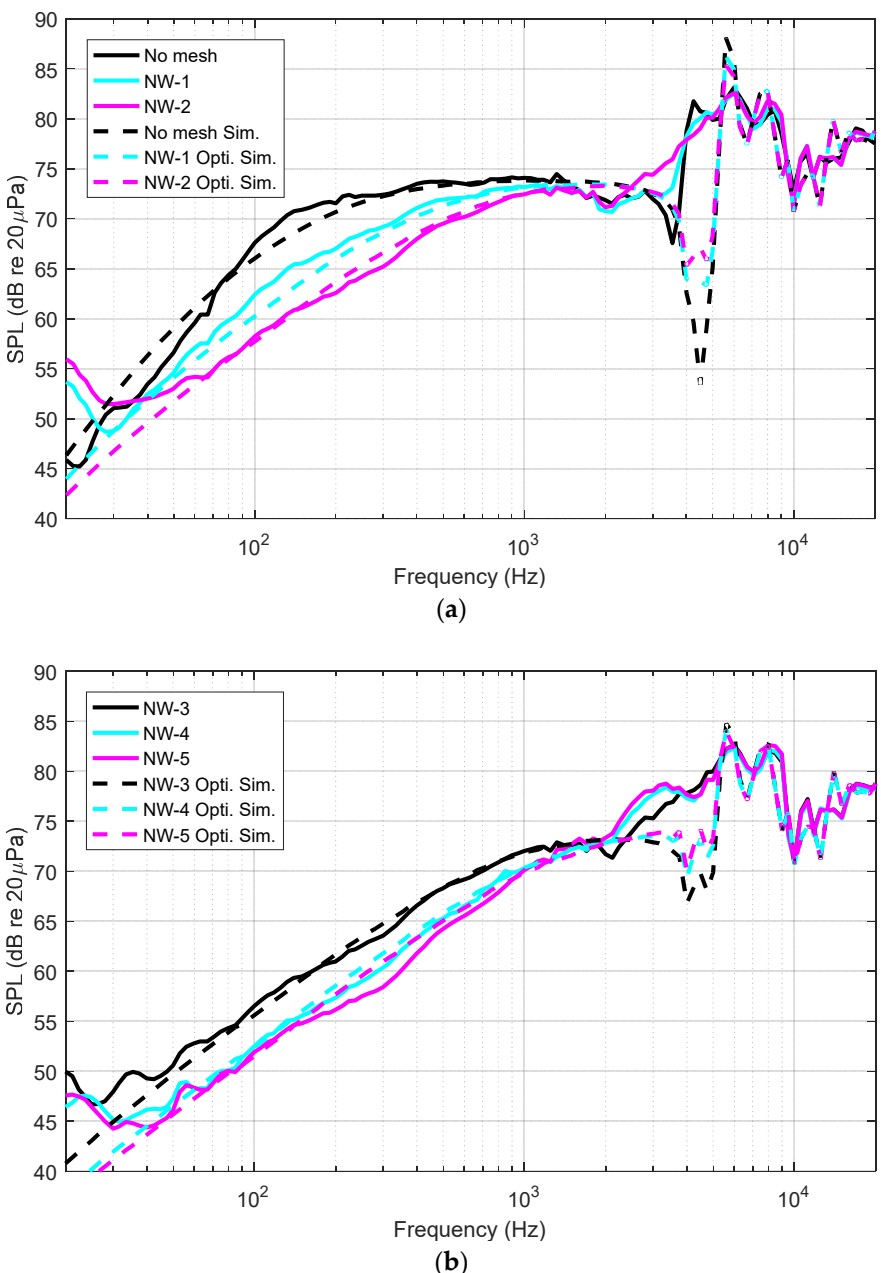

**Figure 10.** (**a**) Measured and simulated frequency responses of microspeaker without and with non-woven ventilation cloths (NW-1 to NW-2) and (**b**) Measured and simulated frequency responses of microspeaker with non-woven ventilation cloths (NW-3 to NW-5).

Furthermore, Figure 11a,b show the comparisons of the measured and simulated frequency responses of microspeaker without and with meshed ventilation cloths (M-250 to M-450). As observed in Figure 11a,b, measured and simulated responses of microspeaker with and without meshed ventilation cloth are satisfactory. However, some variations between measured and simulated responses are seen for NW-1 to NW-5 (Figure 10a,b) at the start of the curve that is due to low frequency (high wavelength) and microspeaker dimensions. The same types of variations were also observed for M-250 to M-450. However, the differences observed at the second resonance (NW-1 to NW-5) might be due to resonance conditions. These observations and their corresponding explanations are also applicable to Figure 11a,b.

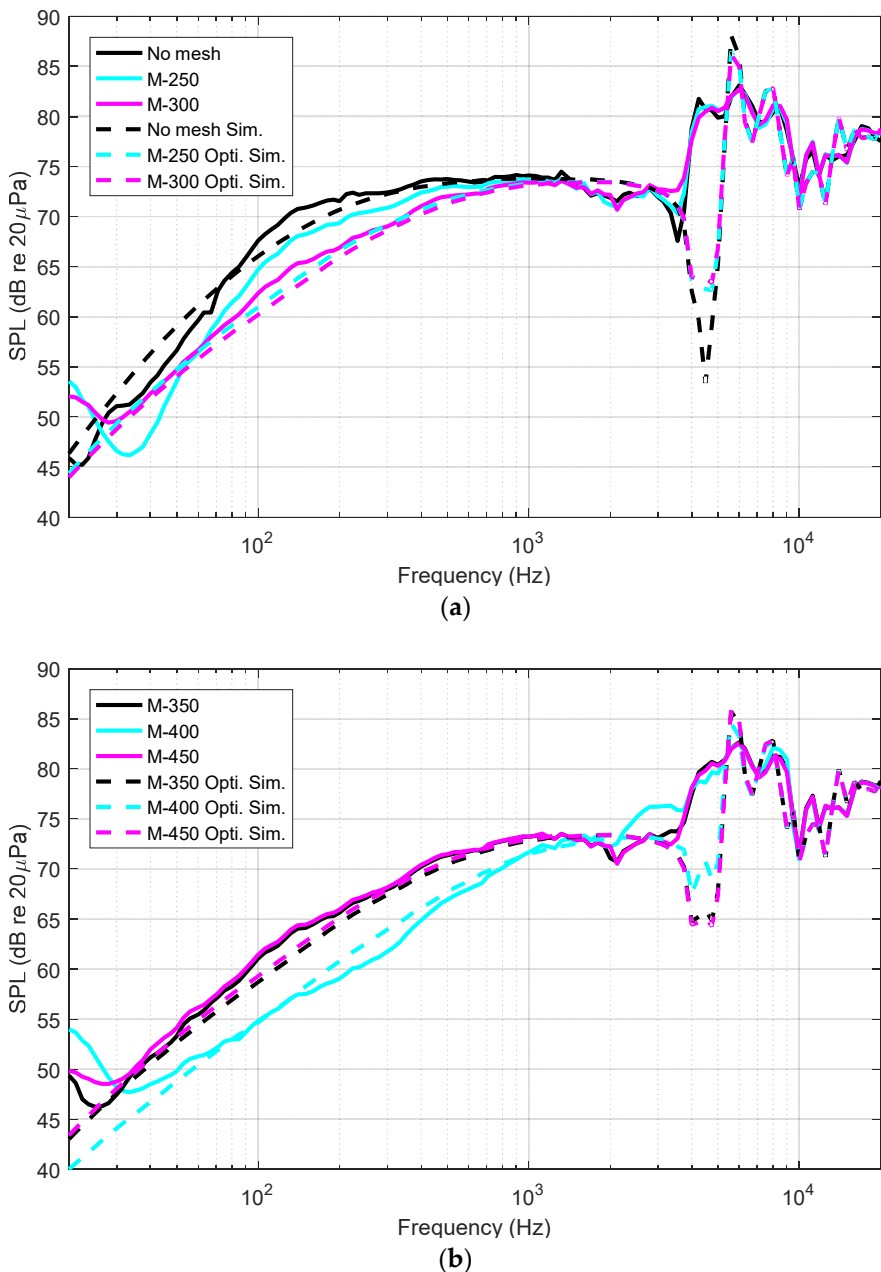

**Figure 11.** (**a**) Measured and simulated frequency responses of microspeaker with meshed ventilation cloths (M-250 to M-300) and (**b**) Measured and simulated frequency responses of microspeaker with meshed ventilation cloths (M-350 to M-450).

## 7. Conclusions

PSO for the inverse calculations of the acoustic impedance of ventilation cloths provided good estimation of the acoustic impedance. The measured frequency response of the microspeaker was successfully used for PSO to produce stable and convergent results. Upper and lower boundaries of 10 to 50 times, number of particles of 20 to 50, and the number of iterations of 50 to 200 have provided acceptable results. When it is placed on the infinite baffle, the equivalent circuit of the microspeaker was successfully formed using the estimated values of the acoustic impedances of 10 common types of ventilation cloths (non-woven and woven). The measured and simulated responses of the microspeaker were found in good agreement. The simulated and measured responses showed varia-

tions in the low-frequency range; however, there was no significant deviation after the fundamental responses.

**Author Contributions:** Conceptualization, J.-H.H.; Methodology, J.-H.H.; Writing—Original Draft Preparation, S.J.P.; Writing—Review and Editing, Y.-C.L. All authors have read and agreed to the published version of the manuscript.

**Funding:** This research was supported by the Ministry of Science and Technology of Taiwan under Contract Nos. MOST 107-2221-E-035-074-MY3 and MOST 108-2218-E-035-007.

**Institutional Review Board Statement:** The study did not require ethical approval.

**Informed Consent Statement:** Not applicable.

**Data Availability Statement:** Not applicable.

**Acknowledgments:** The authors thank Merry Electronic Co. Taichung, Taiwan, and Listen Inc. for supplying the microspeaker and SoundCheck software, respectively.

**Conflicts of Interest:** The authors declare no conflict of interest.

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
