# Peer review of "Inverse Determination of Acoustic Properties of Acoustic Ventilation Cloth (Woven and Non-Woven) by Particle Swarm Optimization and Estimation of Its Effect on the Frequency Response of the Microspeaker"

_applsci, doi:10.3390/app12178588_

Round 1

Reviewer 1 Report

The manuscript applsci-1681149 entitled “Inverse Determination of Acoustic Properties of Acoustic Ventilation Cloth (Woven and Non-Woven) by Particle Swarm Optimization and Estimation of Its Effect on the Frequency Response of the Microspeaker” is focused on experimental measurement and mathematical simulation of frequency dependencies of the sound pressure level (SPL) of the microspeaker type DSH 742-005. The measured and simulated frequency dependencies were determined for the microspeaker that was tested for 10 different types of ventilation cloths (5 non-woven cloths varying in thickness and 5 meshed cloths varying in number of meshes). These characteristics were compared with the frequency characteristics of SPL of the microspeaker without any cloth. The mathematically simulated frequency responses of SPL were obtained using an ECM circuit. In this model, acoustic properties of the studied cloths were obtained from the measured frequency response. Subsequently, the acoustic impedance (or acoustic resistance) of ventilation cloths was inversely determined by particle swarm optimization (PSO) and used in the ECM simulations. Experimental measurements of the frequency characteristics of SPL were performed in an anechoic chamber using a B&K 4191 type microphone. In conclusion, the measured and simulated frequency responses of SPL were evaluated and compared. It can be concluded that the results of this paper are interesting (measured and simulated frequency dependencies of SPL are very similar), but before paper could be published in Applied Sciences MDPI journal, significant improvements must be brought by the authors as following:

  1. In this work, acoustic properties of different types of woven and non-woven cloths on microspeaker are evaluated. But in the discussion chapter (see Figs. 5-8 incl. the manuscript text), the frequency responses are evaluated for papers. Paper and cloth are different materials. What materials do you investigate, papers or cloths?
  2. In terms of the English language, there are many gramatical and formal errors in the manuscript. For example: “cloths (5 nos.) consists of” (Line 20), “criteria has also been established“ (Line 44), “acoustic properties depends” (Line 68), “sound flow limits“ (Line 109), “Total 10 lists different ventilation cloths” (Line 205), “Figure 5 and 6 are measurement and simulation results” etc. The font (Lines 215-229) is too small. Table labels are different (e.g., Table 5 and Table V on page 9). Many other examples can be given. The language quality of his paper is generally poor. Therefore, it is necessary to revise carefully this manuscript by a native speaker or MDPI services.
  3. Line 39: “it became the most significant design on market”. What market? The market for all products?
  4. Line 103: “The resistance of the ventilation cloths (papers) to the airflow changes the amount of air passing through it, hence ventilation cloths can achieve the purpose of adjusting sound quality”. What about optimizing the airflow resistivity of cloths (papers) to obtain a quality sound? Is it better to use materials with lower or higher airflow resistivity? What is the optimum value of the airflow resistivity?
  5. The quality of the figures and Table 1 should be much better, i.e., in terms of their size (e.g., Fig. 1 and 3) and font size (e.g. Fig. 1, 3, 5-8).
  6. Tables 2 and 3 show detailed specifications of the tested cloths. What are the differences of 5 embedded pictures in both tables? There are no differences (e.g. in Table 3 in terms of number of meshes) in the pictures. In my opinion, it is not necessary to insert the pictures into the tables. For the better clarity of the readers, it is much better to insert microscopic photos (of the studied woven and non-woven cloths). It is also appropriate to specify the cloths (manufacturer, types etc.).
  7. Figure 2: It is necessary to describe all quantities (e.g., ZCar1) in the manuscript. Similarly, it is necessary to explain the quantities from Equation (1). The sentence “All symbols have their usual meaning” is not enough.
  8. Figure 3: The B&K 4191 device should be explained. Is the measuring equipment complete? The measuring device is suitable to describe and draw in detail in Fig. 1.
  9. In this paper, frequency characteristics of SPL of the microspeaker are evaluated. It is suitable to define the SPL (equation, significations of quantities) for readers.
  10. Frequency characteristics of SPL were experimentally measured and simulated. The measured and simulated frequency characteristics of the cloths are shown in separate figures. For better clarity, it is useful to compare the measured and simulated characteristics in figures (some examples) to see the differences between the measured and simulated characteristics.
  11. 35: The name of the journal is incorrect.

Summary: It is clear from the above that this manuscript cannot be published in this form until the authors have made thorough and careful revisions.

April 3, 2022.

Reviewer 2 Report

In this study the frequency response of microspeaker structure is investigated. During the measurements the effects of the ventilation cloths are also taken into consideration. PSO algorithm is used for determining the acoustic properties of the ventilation cloths using inverse determination method. The obtained results are new and very interesting because until this time these characteristics of the ventilation cloths were neglected, but their effects on the final results can be important. 

The description of the measurements (chapter 5) and of the inverse calculation of acoustic impedance  (chapter 4) are more or less understandable, however chapter 5 is a littlebit short and extensive litterature research could be necessary for some readers trying to follow and understand some important details. 

In chapter 5, the explaining of the meaning of T/S parameters is missing in the line 231 because the reader is searching the meaning immediately, therefore the sentence explaining the meaning of the T/S parameters should be moved from line 244 into line 231. In line 236 and 237 the text says that figure 3 will show the unechoic chamber too, but this is not so. Please improve the figure 3. In Table 3 the minus sign is missing regarding the numbers in the right column of the table. (ten to minus 5 for example) .

In line 219 instead of warm it would be swarm necessary.  

The description of the PSO process (lines 22-24, 112-116, 193-205, 216-229 and 312-319) is too marginal and too short, it requires an extensive and inconvenient litterature research and investigation if a reader would like to understand or follow the optimization process, even if he is familiar with the PSO or with the optimization. Please give more detailed description about the definition of your optimization problem, answering detailly for the following questions:

  • Exactly what is the objective function (equation) and
  • What are the design variables (how many design variables?)
  • What are the design constraints (explicite and implicite constraints)
  • During the presentation of the optimization results, please describe what are the values of the design variables which give the optimum result of the objective function?
  •  
  • I propose the publication of the paper after giving the answers to these questions. 

Round 2

Reviewer 1 Report

The manuscript applsci-1681149 entitled “Inverse Determination of Acoustic Properties of Acoustic Ventilation Cloth (Woven and Non-Woven) by Particle Swarm Optimization and Estimation of Its Effect on the Frequency Response of the Microspeaker” was reviewed again. It can be concluded that my requirements were supplemented and answered in the work. However, the following revisions need to be made:

  1. Again, there are many grammatical errors in this manuscript in terms of English language, i.e. writing definite and indefinite articles for nouns, commas in sentences, incorrect word bindings (e.g. “a typical three loop circuit” on Lines 197-198, “Finally, the sound pressure level of the microspeaker, which is the pressure level of the sound measured in decibels (dB) and abbreviated as dBSPL” on Lines 205-206 ), singular and plural writing (e.g. “Each loop is driven by velocity v1, v2, and v3, respectively” on Lines 198-199) etc. Furthermore, some sentences are more colloquial (e.g. Figures 6 and 7 are the measurement – see Line 365, Figures 8 and 9 are the measurement – see Line 384, which is 10 cm from the microspeaker – see Line 161, etc.). I could mention a lot of other grammatical shortcomings. Therefore, it is necessary to revise very carefully this manuscript again, in this case by a certified translation agency for the translation of professional English texts.
  2. Similarly, there are formal shortcomings in the manuscript. For example: Line 192 (cavity. Ca), names of Figures 10 and 11 (. and (a) – should be (b)). It is necessary to check it again.
  3. Based on my recommendation, the sound pressure level (SPL) is expressed by the equation (6). Why did you label this quantity as dBSPL? The quantity is marked in Figures 5-11as SPL. And its unit is dB. It can be confusing for readers. I would use only the SPL abbreviation (i.e., without dB) in the equation (6).
  4. Explain this sentence: “This minimum audible level typically occurs between 3-4 kHz“ (see Lines 209-210). In what cases is it valid? In the case of microspeaker? Is it not possible at very high (over 10 kHz) and low (about 20 Hz) frequencies of acoustic waves? It is appropriate to cite the relevant literature. Furthermore, you can compare this statement with the Figures 5-11 (mainly at very low frequencies). Is this valid?

Summary: It is necessary to make major revisions of the manuscript.

May 10, 2022.

Author Response

Reply to reviewer1(Round2) and revised paper are merged to a pdf and attacehed now.
